# Molecular Characterization of an *Isoflavone 2′-Hydroxylase* Gene Revealed Positive Insights into Flavonoid Accumulation and Abiotic Stress Tolerance in Safflower

**DOI:** 10.3390/molecules27228001

**Published:** 2022-11-18

**Authors:** Jianyu Liu, Naveed Ahmad, Yingqi Hong, Meihua Zhu, Shah Zaman, Nan Wang, Na Yao, Xiuming Liu

**Affiliations:** 1Engineering Research Center of the Chinese Ministry of Education for Bioreactor and Pharmaceutical Development, College of Life Sciences, Jilin Agricultural University, Changchun 130118, China; 2Institute of Crop Germplasm Resources, Shandong Provincial Key Laboratory of Crop Genetic Improvement, Ecology and Physiology, Shandong Academy of Agricultural Sciences, Jinan 250100, China; 3Tea Research Institute, Shandong Academy of Agricultural Sciences, Jinan 250100, China

**Keywords:** *CYP81E8*, isoflavone 2′-hydroxylase, constitutive expression, flavonoid biosynthesis, transgenic *Arabidopsis*, safflower

## Abstract

Flavonoids with significant therapeutic properties play an essential role in plant growth, development, and adaptation to various environments. The biosynthetic pathway of flavonoids has long been studied in plants; however, its regulatory mechanism in safflower largely remains unclear. Here, we carried out comprehensive genome-wide identification and functional characterization of a putative *cytochrome P45081E8* gene encoding an isoflavone 2′-hydroxylase from safflower. A total of 15 *CtCYP81E* genes were identified from the safflower genome. Phylogenetic classification and conserved topology of *CtCYP81E* gene structures, protein motifs, and cis-elements elucidated crucial insights into plant growth, development, and stress responses. The diverse expression pattern of *CtCYP81E* genes in four different flowering stages suggested important clues into the regulation of secondary metabolites. Similarly, the variable expression of *CtCYP81E8* during multiple flowering stages further highlighted a strong relationship with metabolite accumulation. Furthermore, the orchestrated link between transcriptional regulation of *CtCYP81E8* and flavonoid accumulation was further validated in the yellow- and red-type safflower. The spatiotemporal expression of *CtCYP81E8* under methyl jasmonate, polyethylene glycol, light, and dark conditions further highlighted its likely significance in abiotic stress adaption. Moreover, the over-expressed transgenic Arabidopsis lines showed enhanced transcript abundance in OE-13 line with approximately eight-fold increased expression. The upregulation of *AtCHS*, *AtF3′H*, and *AtDFR* genes and the detection of several types of flavonoids in the OE-13 transgenic line also provides crucial insights into the potential role of *CtCYP81E8* during flavonoid accumulation. Together, our findings shed light on the fundamental role of *CtCYP81E8* encoding a putative isoflavone 2′-hydroxylase via constitutive expression during flavonoid biosynthesis.

## 1. Introduction

Safflower (*Carthamus tinctorius* L.) is well-known as an economically significant plant species all over the world. Dried safflower petals are a well-documented component of Chinese traditional medicine for the treatment of a wide range of diseases, including heart disease, high blood pressure, gynecological disorders, poor brain circulation, and stroke [1]. Flavonoids, fatty acids, phenolic compounds, and lignin derivatives are some of their other well-known attributes [2]. Many different types of flavonoids were found in safflower, including the chalcone glycoside carthamin, the glucosides of kaempferol, the glucosides of hydroxy safflor yellows A and B, and the glucosides of quercetin [3,4,5,6,7,8]. Safflower’s economic relevance underlines the huge genomic variety for genome-wide associated investigations of gene families involved in flavonoid synthesis [9]. Several attempts have been made to decipher the underlying molecular mechanisms involving fatty acid accumulation [10] and flavonoid metabolism in safflower [11]. Many genes encoding flavonoid biosynthesis enzymes in safflower have already been discovered, including chalcone isomerases (CHIs), chalcone synthases (CHSs), flavanone 3-hydroxylases (F3Hs), and UDP-glucuronosyltransferases (UGTs) [12]. However, the evolution and explication of gene families underlying the molecular mechanisms related to flavonoid biosynthesis remain inexplicit.

Many plant species include hundreds of copies of the cytochrome P450 gene family in their genomes [13]. The study of P450 enzymes has shed light on the intricate interactions between this enzyme’s subunits, the genes it regulates, and the catalytic processes it performs as plants adjust to changing environmental conditions [14,15]. Cytochrome P450 enzymes were mostly identified during the regulation of specialized metabolic pathways such as secondary metabolites, hormones, fatty acid conjugates, and oxidative detoxification pathways in plants [16,17,18]. Isoflavone synthase, also known as 2-hydroxyisoflavanone synthase, is responsible for synthesizing the isoflavonoid backbone from a 4′,7-dihydroxyflavanone substrate (liquiritigenin) via a novel aryl transfer reaction. Multiple legume species, including soybean, may undergo IFS-mediated naringenin conversion at a lower rate [19]. The synthesis of the isoflavonoid skeleton via a putative 2-hydroxyisoflavanone synthase in the licorice plant has previously been reported [20]. Flavanone 2-hydroxylase (F2H) and isoflavone 2′-hydroxylase (I2′H) have previously been identified as members of CYP93B1 and CYP81E1 subfamily and regulate flavonoid biosynthesis in licorice [21,22]. Microsomal extracts from of soybean [23], alfalfa [24], and chickpea [25] have been shown to possess isoflavone 2′ hydroxylase (I2′H) activity, suggesting that this enzyme is a cytochrome P450 family. These studies provide important insights toward the understanding of the crucial role of genes encoding isoflavone biosynthetic enzymes. However, the identification and functional characterization of the CYP81E subfamily in safflower still remained unclear. Thus, it is essential to provide a deep understanding of the genomic diversity of the safflower genome with a particular emphasis on the CYP81E subfamily underlying the molecular mechanism of flavonoid biosynthesis.

The lack of a high-quality reference genome sequence for safflower has significantly hampered the efforts to understand the regulation mechanism of flavonoid biosynthesis. Until now, several attempts have been commenced to provide a full-length genome sequencing of safflower, including one project of full genome sequence using the short-read sequencing method. Similarly, a draft genome sequence of safflower was presented, which consists of a total size of 866 million bp with a single short insert library of ~21× depth [26]. In addition, our previous study also analyzed transcriptomic profiling safflower covering approximately 10.43 GB of clean data, out of which 38,302 sequences were obtained [27]. We also performed the de novo transcriptome assembly of safflower, presenting the identification of oleosin genes [28] and a group of genes involved during safflower yellow biosynthesis [9] using the Solexa-based deep sequencing method. In this study, we report the identification, structural and functional diversity, conserved topology, and diverse expression pattern of CtCYP81E-encoding genes from safflower. In addition, the overexpression of a putative isoflavone 2′-hydroxylase in transgenic *Arabidopsis* provides important insights into the regulation of flavonoid biosynthesis and abiotic stress responses. Together, these findings will highlight crucial hallmarks underpinning the molecular mechanism of isoflavone 2′-hydroxylase-mediated regulation of flavonoid biosynthesis in safflower.

## 2. Results

### 2.1. Identification and Physicochemical Properties CtCYP81E Subfamily in Safflower

Safflower CtCYP81E genes were identified from the genome sequence of safflower selected based on the P450 Pfam00067 domain conserved domain. To further classify the CtCYP81E genes, we performed the hidden Markov model searches (HMM) and used *Arabidopsis* P450 protein sequences as input for the local BLAST search. After removing incomplete transcripts lacking P450 functional domains, a total of 15 full-length CtCYP81E genes were identified. The subfamily members of CtCYP81E were numbered from CtCYP81E1–15. In addition, the physicochemical properties of safflower CtCYP81E proteins were determined, and the results are given in (Table 1). The protein sizes of CtCYP81E encoded amino acids were found in the range of 115–516 amino acids. CtCYP81E proteins were found to have a molecular weight distribution of 12.97–59.01 kDa (CtCYPE9–CtCYPE14), with an average of 45.59 kDa. Isoelectric points (pI) varied from a low of 4.74 (CtCYPE9) to a high of 9.35. (CtCYP81E10). Most of the CtCYP81E proteins were found to have a hydrophilic composition, as indicated by the GRAVY index.

### 2.2. Phylogenetic Analysis of CtCYP81E-Encoding Genes

The phylogenetic tree analysis may be used to determine the origin and evolution of a group of organisms, species, or specific genes. Similarly, phylogenetic trees are helpful for organizing data on biological diversity, constructing taxonomic groups, and shedding light on evolutionary history. We, therefore, utilized the protein sequences of CtCYP81E genes from safflower and other P450s genes in *Arabidopsis* to construct a neighbor-joining (N-J) phylogenetic tree (Appendix A). The P-distance approach was used to divide clusters between these two plant species in the MEGA-X software to establish their evolutionary relationship [29]. A-type and non-A-type P450 sequences are the two primary groups into which P450 genes have typically been categorized. A total of nine clans were grouped that fall within these two clades. These clans include: clan71, clan51, clan710, clan710, clan85, clan711, clan86, clan97, clan72, and clan74 (Figure 1). The CYP71 family was discovered to have the most A-type genes (131), which is 48.51% of the total genes, and has been split into 10 further subfamilies (CYP71AH, CYP71AT, CYP71AU, CYP71AX, CYP71D, CYP71BE, CYP71BG, CYP71BL, CYP71BN, and CYP71BP). Generally, the members of the CYP71 clan represent plant-specific enzymes that play essential roles in plant-specialized metabolite biosynthesis [20,30].

### 2.3. Analysis of CtCYP81E Gene Structure, Protein Motifs, and Cis-Elements

Gene structure, protein motif signatures, and cis-regulatory regions were analyzed for each member of the CtCYP81E family. The results showed that the majority of CtCYP81E genes had a similar exon/intron arrangement (Appendix A). However, in a few CtCYP81E genes, the original intron/exon structure was lost. Some genes, such as CtCYP81E1 and CtCYP81E6, have been shown to have a short exon and a lengthy intron by studying their gene structures. Furthermore, the smallest exon of CtCYP81E6 in safflower (16 bp) was discovered to be longer than the shortest exon of the CYP71 clan (CYP71B32) in *Arabidopsis* (27 bp). There are likely between one and three exons in the safflower CYP81E genes, as shown in (Figure 2a). The majority of these genes had overlapping exonic structures in twos (60%) or threes (26.6%; 4/15) rather than in singles (13.3; 2/15). In contrast, safflower P450 genes have intron lengths that are consistent with those seen in A. thaliana, ranging from 126 to 5946 base pairs (bp) [31] and C. elegans CYPs [32]. Similar to other members of the P450 family, CtCYP81E-encoding proteins possessed typical hallmark features such as the heme-binding domain, the peptide-exchange and repression factor (PERF) domain, the K-helix domain, and the I-helix motif (Appendix A). In addition, a MEME search turned up ten conserved motifs shared by CtCYP81E proteins. Inferences from the data suggested that essentially all putative CtCYP81E proteins possessed these conserved patterns, with the exception of CtCYP81E4, CtCYP81E9, and CtCYP81E11. Based on these findings, it appears that basic structural and functional domains are carried down from one generation to the next in the evolution of CtCYP81Es in safflower (Figure 2b).

Additionally, we investigated the cis-regulatory components of CtCYP81Es encoding genes conserved in the 2 kb region of the promoter sequence (Appendix A). In the 2 kb 5′flanking region of the promoter, there were six major types of cis-elements observed (Figure 2c). Some of the cis-elements that are abundantly expressed with a great number are light-responsive units (G-box; AAAC-motif) and hormone-responsive units (methyl jasmonate, or MeJA). In addition, some genes displayed the occurrence of endosperm expression units (AACA motif; GCN4 motif), low-temperature responsive units (P-box; TATC-box), abscisic acid (ABA) responsive units (ABRE), and anaerobic induction units (ARE). The presence of these well-known cis-elements provided conclusive evidence that CtCYP81Es are strongly associated with plant growth, development, adaptability to diverse stressors, and other essential signaling pathways.

### 2.4. Protein Interaction Network Prediction

Specialized substances, including flavonoids, vitamins, steroids, hormones, and fatty acids, are synthesized via the xenobiotic pathway, in which the monooxygenases (cytochrome P450) play crucial roles. However, P450-encoded enzymes require the interaction of their redox counterparts in order to catalyze substrate-specific biochemical reactions in plants. We, therefore, investigated the protein–protein interaction (PPI) network of the CtCYP81E members by using AtCYP81E orthologs in the STRING database. Various interactor proteins, including translocation, membrane lipoprotein, aquaporin-like, ABC transporter, and protein kinase, are co-associated with the AtCYP81E (Figure 3). Based on these results, we hypothesized that CtCYP81E protein (1, 2, 3, 5, 6, 7, 9, and 15) would also be related to the unique interactor proteins of the UGT74E2 protein, which plays a major role in the production of IBA (indole-3-butyric acid) and significantly affects auxin homeostasis. In the same way, proteins AT5G25930, which are mostly involved in phosphorylating amino acids in proteins, collaborate with CtCYP81E proteins (1, 2, 3, 5, 6, 7, 9, and 15) and CtCYP81E4-8-11 forming complexes with the AT5G48605 protein to improve plant resistance. CtCYP81E protein (4, 8, and 11) also links with AT1G59660, a critical regulator in the water channel. We observed that proteins other than CtCYP81E (4, 8, and 11) interact with ABCD1 proteins, suggesting that they may play a role in the specialized reactions. The protein–protein interaction (PPI) network of CYP81E-encoding proteins suggests many functions for this gene in plant physiology and biosynthesis.

### 2.5. Tissue-Specific Expression of CtCYP81E-Encoding Genes in Safflower

Using RNA-seq (whole-Transcriptome Shotgun Sequencing), we first analyzed the expression patterns of CtCYP81E genes in five distinct tissues/organs of safflower (root, stem, seed, flower, and leaf tissues). The differential expression of CtCYP81E genes in safflower leaves, stems, seeds, flowers, and roots is depicted in Figure 4a, where the genes are grouped into five categories ranging from G1–G5. From the interpretation of RNA-seq data, we further chose the flower tissue of safflower, indicating the highest percentage of differentially expressed CtCYP81E genes to investigate their regulatory mechanism underlying flavonoid biosynthesis. To validate the integrity of the transcriptomic data and differential expression of *CtCYP81E* genes, we analyzed 15 genes by quantitative reverse transcription-polymerase chain reaction during the bud, initial, full, and fading phases of flowering. CtCYP81E2, CtCYP81E8, and CtCYP81E15 all showed significant increases in expression during the late blooming stage, suggesting a potential relationship between the expression of CtCYP81E genes and metabolic activity in safflower. In contrast, the patterns of CtCYP81E1, CtCYP81E2, CtCYP81E5, and CtCYP81E7 expression were upregulated during the fading phase. Similarly, the expression level of *CtCYP81E14* and CtCYP81E15 showed upregulation at the initial flowering stage (Figure 4b). Together, these results suggested that the differential expression pattern of *CtCYP81E* during flower developmental phases might correlate with secondary metabolism in safflower.

### 2.6. Differential Expression of CtCYP81E8 Encoding a Putative Isoflavone 2′-Hydroxylase Correlates with Metabolite Accumulation in Different Flowering Stages

The expression analysis of *CtCYP81E8* encoding a putative isoflavone 2′-hydroxylase gene in different flowering stages was investigated along with the accumulation of total metabolite content was performed using qRT-PCR analysis and quantification of total metabolite content. The expression analysis of CtCYP81E8 through different flower developmental stages showed a consistent pattern with the accumulation of total metabolite content. As shown in (Figure 5a,b), the expression level of CtCYP81E8 at the initial flowering stage was slightly increased, and at the same time, the total metabolite content was also increased at the same time when compared to the bud flowering stage. Similarly, the expression level of CtCYP81E8 at the full flowering stage reached its maximum, and at the same time, the total metabolite content also increased significantly compared to the flower bud stage. Importantly, when the expression level of CtCYP81E8 at the fading stage declined, the metabolite content also decreased. These results unanimously indicated a strong relationship between CtCYP81E8 (isoflavone 2′-hydroxylases) expression and metabolite accumulation through different flowering stages in safflower.

### 2.7. The Orchestrated Link between Transcriptional Regulation of CtCYP81E8 Encoding a Putative Isoflavone 2′-Hydroxylase and Flavonoid Accumulation in the Yellow and Red Flowering of Safflower

The relationship between the expression of isoflavone 2’-hydroxylase at the transcription level and flavonoid accumulation was further investigated in the two varieties of safflower that bloom naturally (red and yellow) (Figure 6a,b). Interestingly, the expression pattern of *CtCYP81E8* showed a programmed pattern of expression correspondingly with the accumulation of flavonoids during multiple flowering stages in yellow-type flowers. The expression level of *CtCYP81E8* at the Y1 flowering stage (bud) showed no significant effect on the accumulation level of flavonoid content in yellow-type flowering. However, at the Y2 (initial flowering stage), the relative expression of *CtCYP81E8* demonstrated an increased expression level with an abundant increase in the accumulation level of flavonoids (Figure 6c). Similarly, the expression level of *CtCYP81E8* at the Y3 (full) and Y4 (fade) stages showed a relatively high expression compared to the Y1 (bud) stage; however, the accumulation of flavonoid content in both of these stages was significantly increased (Figure 6c).

On the other hand, the correlation of *CtCYP81E8* expression and metabolite accumulation level through different stages of red-type flower showed a diverse transcription regulation pattern. Our findings suggested that the expression level of *CtCYP81E8* at the R1 (bud) and R2 (initial) stages showed a discontinued pattern, resulting in the increased expression level and decreased metabolite accumulation (Figure 6d). Noticeably, the expression level of *CtCYP81E8* at the R3 (full) stage was significantly increased compared with the earlier stages, whereas the flavonoid accumulation level reached its maximum at the R3 flowering stages. The regulation of *CtCYP81E8* transcription at the R4 (fade) stage showed an opposite trend resulting in decreased expression level and increased flavonoid accumulation. Based on these findings, we speculated that the transcriptional regulation of *CtCYP81E8* might be linked to the accumulation of flavonoids in safflower. These results may play a key role in comprehending the fundamental idea of the CtCYP81E8-induced genomic regulatory mechanism, which purposefully switches on metabolic flux by altering its transcriptional expression.

### 2.8. Expression Profiling of CtCYP81E8 Encoding a Putative Isoflavone 2′-Hydroxylase under Different Abiotic Stress Conditions

The expression level of CtCYP81E8 was investigated under variable stress conditions using qRT-PCR assays. The expression pattern of CtCYP81E8 under methyl jasmonate stress showed a differential expression at different time points (0–60 h). The expression of CtCYP81E8 showed an increased pattern at 48 h, at which the transcription level reached to its maximum compared with the control group. However, the expression level of CtCYP81E8 was downregulated at the 60 h time point (Figure 7a). The expression of the CtCYP81E8 gene was considerably higher in the drought-stressed plants after 24 to 48 h. Evidently, CtCYP81E8 expression peaked at the 48 h time point. At 60 h of treatment time, however, CtCYP81E8 transcription was suppressed (Figure 7b). Surprisingly, CtCYP81E8 gene expression was lower in the intense light-exposed plants (12–60 h) compared to the control plants. This downregulation primarily occurred after 36 h of treatment, which suggests extreme sensitivity to light stress (Figure 7c). Furthermore, the expression level of CtCYP81E8 under dark conditions mostly indicated upregulation at each time point. However, the expression level w reached its maximum at the 36 h time point under dark conditions. Exceptionally, the expression level of CtCYP81E8 was consistently upregulated at 12–36 h; however, the downregulation of CtCYP81E8 transcription was observed right after 48 h, compared to the control group (Figure 7d). These results highlight important clues towards the possible role of CtCYP81E8 during the plant’s adaptation to abiotic stress environments.

### 2.9. Expression Analysis of CtCYP81E8 Encoding a Putative Isoflavone 2′-Hydroxylase and Other Key Flavonoid Biosynthetic Genes in Transgenic Arabidopsis

The expression analysis of the *CtCYP81E8* transcript in fifteen independent transgenic lines was investigated using the qRT-PCR assay. As shown in Figure 8, the transcript abundance of CtCYP81E8 demonstrated a diverse expression pattern considering fifteen different over-expressed transgenic lines. The highest expression was detected in the OE-13 line, with an approximately eight-fold increase in expression. Similarly, the expression trend of CtCYP81E8 was four-fold upregulated in the OE-5 transgenic line, followed by a three-fold increase in OE-1, OE-7, OE-11, OE-12, and OE-15 transgenic lines (Figure 8). However, the transcription of CtCYP81E8 was slightly upregulated in OE-6, OE-8, OE-10, and OE-14 transgenic lines compared to other lines. Noticeably, the expression level of CtCYP81E8 was not induced in OE-2, OE-3, OE4, and OE-9 transgenic lines. Furthermore, the expression level of ten core structural genes involved in the downstream regulation of the flavonoid pathway of the OE13 transgenic line was further investigated using qRT-PCR analysis. The expression level of *AtANR*, *At4CL*, *AtFLS*, *AtF3H*, *AtCHI*, and *AtLAR* genes was downregulated in the OE-13 transgenic line compared to the wild type. On the contrary, the expression level of the *AtCHS*, *AtF3′H*, and *AtDFR* genes was noticeably upregulated in the OE-13 line than the wild type. Together, our findings suggested that the increased expression level of CtCYP81E8 in transgenic plants particularly induces the expression of the upstream flavonoid biosynthetic genes and, therefore, could enhance the accumulation level of flavonoids in safflower.

### 2.10. Determination of Flavonoid Contents in CtCYP81E8 Overexpressed Transgenic Arabidopsis Using HPLC-MS/MS

The conventional chromatographic column method was used for the separation of flavonoid metabolites in the OE-13 transgenic line and wild-type plants. As shown in Figure 9, we detected several types of flavonoid content in the OE-13 transgenic line using HPLC-MS/MS analysis. For instance, the accumulation of dihydroquercetin peak was observed in the OE-13 line compared to the wild-type. Similarly, different peaks representing the presence of kaempferol, naringin, and chalcones in the OE-13 line were also detected when compared to the wild-type. In contrast, epicatechin and rutin were not detected in both the wild-type and OE-13 line. Given these results, we hypothesized that the overexpression of CtCYP81E8 may result in a noticeable change in the total amount of flavonoids in the transgenic plant by inducing the expression of *CHS*, *F3′H*, and *DFR* genes during the regulation of the flavonoid pathway.

## 3. Discussion

Safflower is a significant plant used for many purposes, such as ornamental, food, textile, and medicinal purposes. Based on the growing demand for edible safflower oil and its extensive medical benefits, many research ventures have been performed using whole omics technologies worldwide. By integrating different omics strategies such as genomics, transcriptomics, proteomics, and metabolomics, it is possible to achieve a comprehensive understanding of the molecular and biochemical networks underlying specialized metabolism in safflower [33]. In the same way, we attempted to unveil a functionally diverse class of the enzyme-encoding gene family of Cytochrome P450 (CtCYP81E) concerning flavonoid biosynthesis in safflower. Our genome-wide identification of the CtCYP81E subfamily resulted in a total of 15 putative members in the safflower genome. The phylogenetic classification with *Arabidopsis* P450s suggested clustering all 15 CtCYP81Es to the largest A-type CYP71 based on the amino acid homology level. Our hypothesis was also corroborated by this categorization, suggesting that the majority of CYP71 subfamilies constitute the subset of enzymes participating in important metabolites biosynthesis. With the addition of the new safflower CtCYP81Es, the CYP71 group was found to be the largest A-type class, consisting of 131 genes (48.51%) that were successfully divided into nine subclades; nevertheless, 10 other clans, including CYP71AH, CYP71AT, CYP71AU, CYP71AX, CYP71D, CYP71BE CYP71BG, CYP71BL (Figure 1). Based on our phylogenetic analyses, we know that clans 51, 710, 711, and 74 all pertain to a single family of P450 genes; however, neither of the CtCYP81E-encoding genes was allocated to any of the other five clans [34,35].

The pattern of intron–exon structures and the process of gain and loss mutations significantly highlight the mechanism of evolution of certain gene families that are present in the same phylogenetic clade. Understanding the conserved introns organization offers ancient elements to understand the similar group of genes involved in multiple physiological processes of plants [36]. Our study discovered that the CtCYP81E subfamily genes share similarities with the genes located on clan 71 from other plant genomes, with most genes having between one and three exons and two introns [31,37] (Figure 2a). Furthermore, the conserved residues introns were not discovered in the non-A type *Arabidopsis* P450 gene families comparable to A-type clan71 gene families, showing a major process of intron evolution during gene family assembly [38,39,40]. Five well-known P450 motifs, including a heme-binding motif (PFxxGxRxCxG/A), a C-helix (WxxxR), a PERF motif (PxxFxPE/DR), a K-helix (ExLR), and an I-helix (GxE/DTT/S), were all found in the safflower CtCYP81E subfamily [41,42]. The C-helix motif was shown to have widespread conservation of a small set of amino acid residues, including tryptophan (W) and arginine (R). Similarly, the amino acid residues of glycine (G) and threonine (T) were mostly conserved in the I-helix motif, whereas the amino acid residues of phenylalanine (F), glycine (G), arginine (R), and cysteine (C) were mostly found in heme-binding motif. The K-helix region contains the presence of glutamic acid (E) and arginine (R), while the PERF region consists of the amino acid residues of proline (P). All of the aforesaid conserved residues were detected throughout the CtCYP81E subfamily members from the safflower genome (Figure 2b). Hormonal-responsive elements (methyl jasmonate (MeJA) and C-repeat), Abscisic acid-responsive elements (ABRE), low-temperature stress elements (P-box; TATC-box), dehydration/drought-responsive elements (DRE), and other well-known stress-responsive units were predicted to be present in CtCYP81E-promoters based on the cis-elements conservation and distribution [43,44] (Figure 2c). Subsequently, the unique evolutionary pattern of CtCYP81E in safflower was revealed by a comprehensive survey of its gene structure compositions, conserved motifs, GO classification, and the allocation of widely used cis-regulatory units. This further highlights the functional interplay of these putative genes during plant tolerance to multiple stress reactions and other important functional pathways.

The transcription level of gene expression is directly dependent on the plant growth and development phases, age, environmental factors, tissue specificity, and conflicting responses to multiple biotic and abiotic stresses. In this study, we also evaluated the transcription of putative *CtCYP81E* genes in different flowering stages of safflower. The results from RNA-seq suggested a differential expression pattern of the CtCYP81E genes in safflower by dividing them into five distinct groups among various tissues/organs such as leaves, stem, seed, flowers, and root, respectively (Figure 4a). These findings corroborated a prior study that found that 31.33 percent of P450 genes in *Solanum lycopersicum* exhibit differential expression profiles across many tissues and organs during development. Our results were also in line with the previous findings in soybean (31.92%) [45], mulberry (23.6%) [37], and rice (49.81%) [46].

In addition, the transcriptional control of *CtCYP81E8* in various safflower flowering phases under both normal and stressful conditions was also analyzed. Our findings vividly illustrate the temporal control of *CtCYP81E8* transcript abundance in response to hormonal (MeJA), drought, light, and dark stress conditions (Figure 6). Similarly, the expression level of *CtCYP81E8* in transgenic *Arabidopsis* showed that the higher levels of *CtCYP81E8* transcription might increase flavonoid accumulation by upregulating the expression of upstream flavonoid biosynthesis genes in the pathway. These results significantly highlighted the importance of the complex and periodic regulation network of the *CtCYP81E8* gene under varying climate conditions. Our most recent work on *CtCYP82G24* provided strong support for these conclusions by outlining the fundamental notion of correlating quantitative gene expression trends with metabolite accumulation patterns in response to abiotic stress in transgenic *Arabidopsis* [47]. Further, we found that *CtCYP82C1* transcription was increased in response to both hormonal and abiotic stress [48]. These findings provide a practical basis for the functional characterization of *CtCYP81E8* with an emphasis on flavonoid biosynthesis. However, additional rigorous methodologies are needed to select viable candidate genes from a broad array of the cytochrome P450 supergene family in safflower.

The P450 superfamily is the biggest group of monooxygenases in all known species. Furthermore, the overall trend of this multigene family in plants centers on the unique metabolic pathways that occur at various phases of plant growth. The shikimate biosynthesis route has so far been the principal focus for the functional characterization of the CYP71 clan subfamilies [49,50,51]. All P450s use a combined oxidative and reductive mechanism to catalyze their substrates. However, a tighter relationship between such genes and metabolic remodeling remains challenging. In this study, we focused on the expression of the key genes involved in flavonoid biosynthesis using the CtCYP81E8 overexpressed transgenic line (OE-13) and wild-type in order to illustrate the relationship between gene expression and the accumulation of flavonoids content. Our findings showed that the expression of most of the flavonoid pathway genes, such as *AtANR*, *At4CL*, *AtFLS*, *AtF3H*, *AtCHI*, and *AtLAR*, showed downregulation in the transgenic line compared to the wild-type plant. In contrast, the expression level of *AtCHS*, *AtF3′H*, and *AtDFR* genes was noticeably upregulated in the OE-13 line than the wild-type. These results importantly suggested that the differential expression of key structural genes involved in the flavonoid biosynthetic pathway genes shed light on the possible role of *CtCYP81E8* in flavonoid accumulation. We further validated the HPLC-MS/MS analysis of the OE-13 transgenic plant, which showed that the content of different types of flavonoids such as dihydroquercetin, kaempferol, naringin, and chalcones in the OE-13 plant was detected compared to the wild-type plant. From these findings, we suggested that the overexpression of CtCYP81E8 may cause a significant shift in flavonoid accumulation level. The detection of flavonoid content in the OE-13 transgenic line might correlate with the upregulation of *CHS*, *F3′H*, and *DFR* genes. Our results were consistent with the previous findings suggesting that the synthesis of coumaryl-CoA catalyzed by CHS enzyme is reduced due to the decreased expression level and reduced substrate level [52,53,54]. In the same way, the *CtCYP81E8* gene demonstrated a positive regulatory effect on the *DFR* gene, which may participate in the synthesis of leucocyanidin. Altogether, our findings revealed the partial role of CtCYP81E8 during flavonoid accumulation by upregulating the expression level of other key biosynthetic genes involved in the downstream regulatory pathway of flavonoid biosynthesis in safflower.

## 4. Materials and Methods

### 4.1. Identification and Phylogenetic Analysis of CtCYP81E Subfamily in Safflower

We performed the first genome assembly for Safflower (variety Jihong01), integrating Illumina and PacBio sequencing. The assembled genome of approximately 1.04 Gb consisted of 32,353 protein-coding genes, of which 94.69% were functionally annotated. The members of genes encoding CtCYP81Es were identified from the safflower genome. In addition, the *Arabidopsis* P450 sequences were obtained from the TAIR website (https://www.Arabidopsis.org/, accessed on 25 September 2021) for comparison. The identified CtCYP81Es from the safflower genome were validated by searching the Pfam database (http://pfam.xfam.org, accessed on 25 September 2021) for further verification. Additionally, the online webserver of ExPASy (http://www.expasy.org/, accessed on 25 September 2021) was used to calculate a number of physicochemical characteristics, such as molecular weight (MW) and the isoelectric point (pI) of each discovered CtCYP81E protein. Similarly, subcellular localization was determined using the online server of CELLO v2.5 (http://cello.life.nctu.edu.tw/, accessed on 25 September 2021). Using the preset parameters, the DNAMAN software (Vers7; Lynnon Corporation, Montreal, QC, Canada) was utilized for multiple sequence alignment of CtCYP81E amino acid sequences. Finally, phylogenetic analysis was constructed by employing the neighbor-joining phylogenetic tree using the method of 1000 bootstrap in MEGA 5 software version 4.1 (http://www.megasoftware.net/, accessed on 25 September 2021) following the instructions of [55]. A total of 260 sequences were clustered together in the phylogeny analysis, including 245 P450 sequences from *Arabidopsis* and 15 CtCYP81Es obtained from the Safflower genome.

### 4.2. Gene Structure, Protein Motifs, Promoter Analysis and Protein–Protein Interaction

The CtCYP81E gene CDs and genomic sequences were used with the application of GSDS (Gene Structure Display Server) (http://gsds.cbi.pku.edu.cn/index.php, accessed on 25 September 2021) to track the exon-intron distribution and other aspects of gene structure as described by [56]. The conserved protein motifs were determined by adding the protein sequences of CtCYP81Es to MEME web server Version 4.8.1; available at http://meme.nbcr.net/meme/cgi-bin/meme.cgi, accessed on 25 September 2021) with the default parameters. Protein motif visualizations in the ML Phylogenetic tree were modified in EvolView v.2 (http://www.evolgenius.info/, accessed on 25 September 2021). The online webserver of PLACE with default settings was used to analyze the 2 kb upstream 5’ UTR flanking sequence of each CtCYP81E gene in order to examine the cis-regulatory units of the promoter region. We used the online tool STRING network (https://string-db.org/, accessed on 25 September 2021) to predict the functional protein interaction network of the CtCYP81E proteins (https://string-db.org/). A diverse collection of experimental and hypothetical proteins was predicted to interact with CtCYP81E proteins by using the hierarchical network of interactor proteins. This interaction occurred at both the upstream and the downstream stages of regulation. The PPI information representing both transient and stable interactions of CtCYP81Es was graphically recorded in the STRING database.

### 4.3. Experimental Materials, Vectors, and Strains

The seeds of the “Jihong No.1” cultivar was acquired from Tacheng company, Xinjiang, China. The seeds were cultivated in the artificial growth chamber at the Jilin Agricultural University, Changchun, China, at a temperature of 23 ± 2 °C until they were harvested. The competent cells of *Agrobacterium tumefacien* (EHA105) and the cells of *E. coli* (BL21 and DH5α) were obtained from the Takara Biomedical Technology Co. (Beijing, China).

### 4.4. Differential Expression of CtCYP81E-Encoding Genes in Safflower

Genes encoding CtCYP81E in safflower were analyzed for differential expression using RNA-seq data from roots, stems, seeds, flowers, and leaves. Differentially expressed CtCYP81E genes were given a score based on their FPKM values from RNA-seq data. In addition, the experimental validation of the RNA-seq data was conducted using qRT-PCR analysis to verify the real-time expression of CtCYP81E genes across all flowering phases. The safflower cultivar Jihong No.1 was grown for four months, and the total RNA was isolated from its four distinct flowering stages with the help of an RNA Isoplus reagent kit (TIANGEN Biotech, Beijing, China). The reverse transcription system was used to synthesize first-strand cDNA templates with the use of the PrimeScript™ RT Reagent Kit and gDNA Eraser (TaKaRa, Beijing, China). Then, SYBR^®^ Premix Ex Taq™ (TaKaRa) was used in a quantitative real-time polymerase chain reaction test to assess the relative expression of CtCYP81E-encoding genes. The qRT-PCR analysis was determined using a Stratagene Mx3000P system (Stratagene, San Diego, CA, USA). In order to standardize the results, the safflower 18s ribosomal RNA gene (GenBank accession: AY703484.1) was used to measure the expression level. The 2^−ΔΔCt^ method was used to determine the relative expression level of CtCYP81Es in each flowering tissue [57]. Three biologically identical sets of data were collected for each trial. The primers are listed in (Appendix A).

### 4.5. CtCYP81E8 Expression and Metabolite Accumulation in Different Flowering Stages

In order to investigate the expression pattern of the CtCYP81E8 gene and flavonoid metabolites, we performed an overlapping experiment of examining the expression level of CtCYP81E8 and flavonoid metabolite accumulation in four different flowering stages of safflower. The qRT-PCR analysis of CtCYP81E8 was performed with the forward and reverse primers CtCYP81RT-F (5′TGTATCGCCCACACGTTCACT3′) and CtCYP81RT-R (5′TTTCCGGCAGGTCCTTTGTT3′). Reactions in a quantitative real-time polymerase chain reaction (qRT-PCR) were carried out with the following temperature cycle: one cycle of 5 min at 95 °C, followed by 35 cycles of 10 s at 95 °C, 10 s at 58 °C, and 30 s at 72 °C. We used the safflower 18s ribosomal RNA gene to use as a reference point. The expression data were calculated according to the 2^−ΔΔCt^ method. Three independent biological samples were used for each quantitative investigation. In addition, the total metabolite content in different flowering stages was quantified according to the aluminum chloride colorimetric method.

Similarly, the correlation between CtCYP81E8 expression and the accumulation of total metabolite content in red- and yellow-typed flowers was also investigated. Red and yellow safflower petals were obtained at the bud, initial, full, and fade phases of flower development. The isolation of the total RNA and extraction of total metabolite content were performed simultaneously according to the previous method. The CtCYP81E8 transcription level was analyzed for each flowering type and developmental phase using qRT-PCR assays. Three replicates per development stage were performed in each experiment, and the data were analyzed using the 2^−ΔΔCt^ method. As a reference for comparison, we employed the 18s ribosomal RNA gene from the safflower genome. At the same time, a 14 mL water–alcoholic solution was used to ultrasonically separate the metabolites from the remaining homogenous mixture of flower petals, which was subjected to the standard conditions of 60 °C extraction temperature, followed with 30 min of extraction time twice, and a 10 min centrifugation cycle at 5000 rpm. Finally, a 0.5 mL (1 mg/mL) sample was mixed with 10% aluminum chloride, 1 M potassium acetate, and 80% methanol solution. Spectrophotometric absorbance readings were taken at a wavelength of 415 nm. We used milligrams of total flavonoid content (TFC) per 100 g of fresh weight or dry weight to get the TFC% [58]. To reduce the likelihood of making a mistake, we employed three biological replicates (*n* = 3).

### 4.6. Plant Material and Stress Conditions

Under carefully managed photoperiodic conditions, the safflower seeds were cultivated in an artificially controlled room at Jilin Agricultural University (Changchun 130118, China), College of Life Sciences, Engineering Research Center of the Chinese Ministry of Education for Bioreactor and Pharmaceutical Development. Methyl jasmonate 300 mmol/L was sprayed on safflower plants that had been flowering for 3 weeks; we set our sample times for 0 h, 12 h, 24 h, 36 h, 48 h, and 60 h to simulate various stress situations. In the same way, a 20% PEG-6000 was used to induce drought stress across the same 12-60 h intervals; each 12 h represents a temporal gradient. In both the dark and the light treatment groups, the seedlings were cultivated in a dark box to protect them from any possible exposure to light. At each time point, fresh samples were taken and preserved in liquid nitrogen at −80 °C until future use. Across a range of time points and stress levels, total RNA was isolated from all experimental groups. Using a reverse-transcription procedure, we synthesized first-strand cDNA. CtCYPE8 transcriptional regulation was studied utilizing quantitative real-time polymerase chain reaction (qRT-PCR) analysis using cDNA templates generated by reverse transcription.

### 4.7. Generation of CtCYP81E8 Overexpressed Transgenic Arabidopsis Lines

The full-length cDNA sequence of the CtCYP81E8 gene was amplified from the flower petals of the JH1 cultivar using the forward and reverse primers (CtCYP81-F1-5′CCCATGGGATGATGAGGATGATTAGTGG3′) and CtCYP81-R1-5′ TCAAAGATGCGATAATAGATTTGACTAGTC3′) containing restriction sites of EcoRI (GAATTC) and BamHI (GGATCC), respectively. The sequence of CtCYP81E8 was deposited into the public database of NCBI, available under the accession number (MW436617). The complete cDNA fragment of CtCYP81E8 was transformed into the plant overexpression vector pCAMBIA-3301 under the control of the 35S (CaMV) promoter using T4 ligase. After confirmation with Sanger sequencing, the recombinant vector of pCAMBIA-3301 containing CtCYP81E8 was inserted into A. tumefaciens (EHA105 strain) following the heat and shock method. The empty vector of pCAMBIA-3301 was also transformed into A. tumefaciens (EHA105 strain) to be used as a positive control. The floral dip infiltration method was performed to achieve the genetic transformation of wild-type A. thaliana with CtCYP81E8 overexpression vector following the instructions of [28]. Transgenic plants were further screened out using BASTA spray and PCR amplification to set T2 and T3 generations, respectively [59].

### 4.8. Expression Analysis of CtCYP81E8 and Other Key Flavonoid Biosynthetic Genes in Transgenic Lines

To further explore the expression level of CtCYP81E8 in transgenic lines, we independently selected fifteen OE lines. The total RNA was extracted, and the cDNA templates were prepared for qRT-PCR analysis. The relative expression level of CtCYP81E8 was analyzed in each transgenic line using the 2^−ΔΔCt^ method. In addition, the expression level of ten other key flavonoid biosynthetic genes was also analyzed in the OE-13, which demonstrated the highest expression level of CtCYP81E8. The expression level was compared with wild-type A. thaliana. All qPCR reactions were carried out according to the previously described method. The primer details of key flavonoid biosynthetic genes are given in Appendix A.

### 4.9. Determination of Flavonoids via HPLC-MS/MS Analysis

The flavonoid content was determined in OE-13 and wildtype A. thaliana using HPCL-MS/MS analysis. The conditions during HPCL-MS/MS analysis were as follows: chromatographic column: Agilent SB-C18 (2.1 mm × 50 mm, 2.7 μm); injection volume 10 μL; column temperature at 50 °C, flow rate 0.3 μL/min; mobile phase A is water (5 mmol/ammonium acetate + 0.1% formic acid), B is acetonitrile; the liquid chromatography mobile phase gradient elution procedure is 0 min, 5 min, 10 min, 15 min, and the mobile phase A and B are injected at 10% and 90%, 5% and 95%. Accurately weigh 1 mg of each standard, and dilute to the mark with acetonitrile in a 5 mL volumetric flask. For determination, various types of flavonoid types were considered using different configurations using the standard series of 10, 20, 40, 60, 80, and 100 ng/mL. A sample of 1.0 g fresh leaves was obtained, followed by adding 25 mL (0.1 mol/L HCL) and allowed to mix well, and then centrifuge at 8000 rpm for 5 min. Take 5 mL through the column, and then rinse with methanol for the liquid phase. Finally, the eluent was collected, allowed the evaporation for dryness on a 40 °C rotary evaporator, and then diluted to 2 mL with acetonitrile. The peaks were then analyzed accordingly.

### 4.10. Statistical Analysis

The data were analyzed using the Analysis of variance (ANOVA) method using the SPSS software obtained from three independent replicates. With a *p*-value cutoff of 0.05, the least significant difference (LSD) test was used to compare the means. All graphs were plotted using GraphPad Prism v.8.

## 5. Conclusions

In the present study, we provided comprehensive molecular insights on the genome-wide identification and regulatory mechanism of a putative CtCYP81E8 encoding the isoflavone 2′-hydroxylase gene in safflower. The results demonstrated the potential role of isoflavone 2′-hydroxylase in promoting flavonoid accumulation and abiotic stress tolerance in safflower. In addition, the CtCYP81E8-overexpressed transgenic Arabidopsis uncovered the enhanced expression of several up-regulatory genes involved in the flavonoid pathway. Our findings also confirmed the competitive inhibition of different types of flavonoids in transgenic plants. However, due to the vast functional diversity of cytochrome P450 genes, efforts are still required to fully understand the explicit role of CtCYP81E8 during flavonoid biosynthesis in safflower.

## Figures and Tables

**Figure 1 molecules-27-08001-f001:**
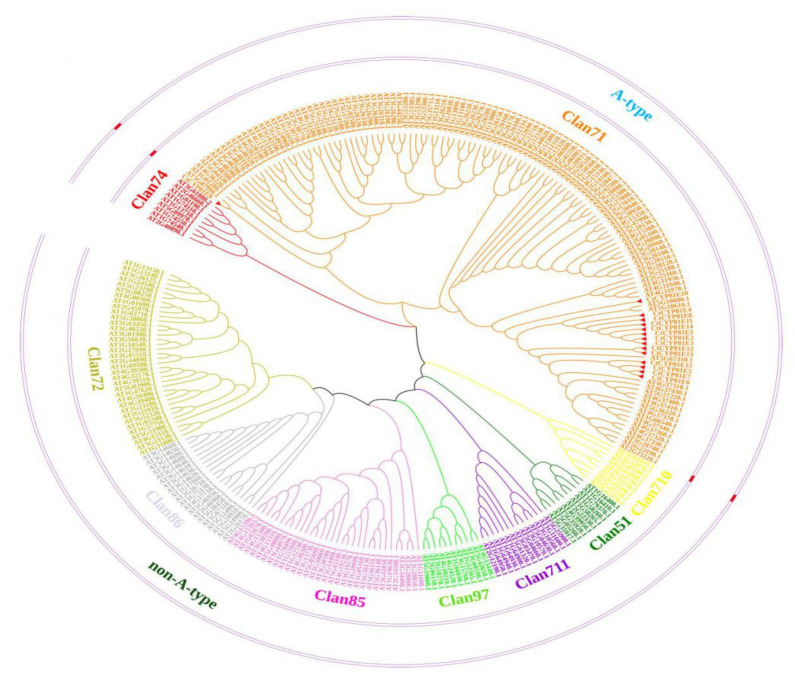
The phylogenetic classification of CtCYP81E-encoding genes with other members of *Arabidopsis* P450s using a neighbor-joining (N-J) phylogenetic tree. MEGA-X software was used to generate the phylogeny tree. The different background colors demonstrated various clans of P450 subfamilies. The CtCYP81E-encoding genes were represented with red triangles in the phylogenetic tree.

**Figure 2 molecules-27-08001-f002:**
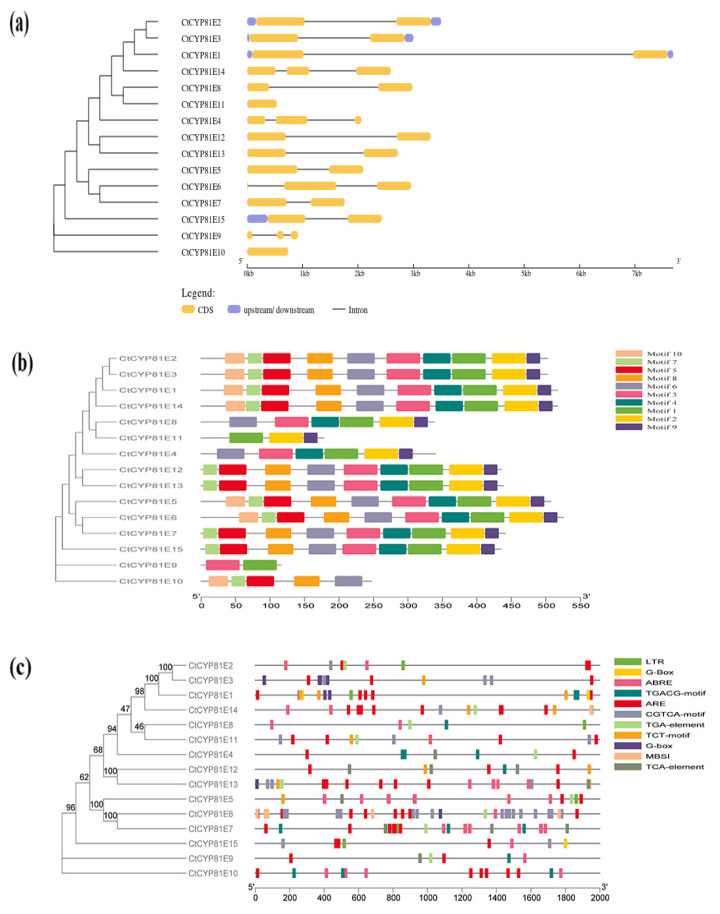
The organization of gene structure, conserved motifs, and promoter analysis of CtCYP81E subfamily members (**a**) The organization of different components of CtCYP81E genes structure such as exon/intron organization and untranslated regions (UTRs). Untranslated regions (UTRs) were shown in blue, exons were depicted in yellow, and the total number of introns was shown in grey (**b**) conserved protein motifs distribution of CtCYP81E encoding proteins. The motif analysis was predicted with MEME webtool (**c**) promoter analysis of CtCYP81E encoding. The presence of different regulatory elements was shown with multiple colors at specific positions.

**Figure 3 molecules-27-08001-f003:**
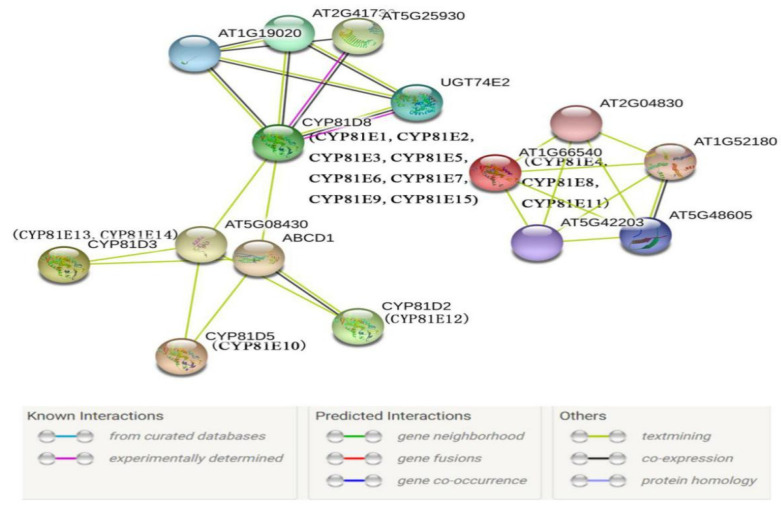
The illustration of the protein–protein interaction network of CtCYP81E encoding proteins. The online webserver of STRING (Search Tool for the Retrieval of Interacting Genes/Proteins) was utilized to annotate the potential network of CtCYP81E8 using the model plant as an input. The red lines represented the presence of more than four different P450 proteins in each set of proteins. Below the *Arabidopsis* protein, in parenthesis, were the CtCYP81E proteins.

**Figure 4 molecules-27-08001-f004:**
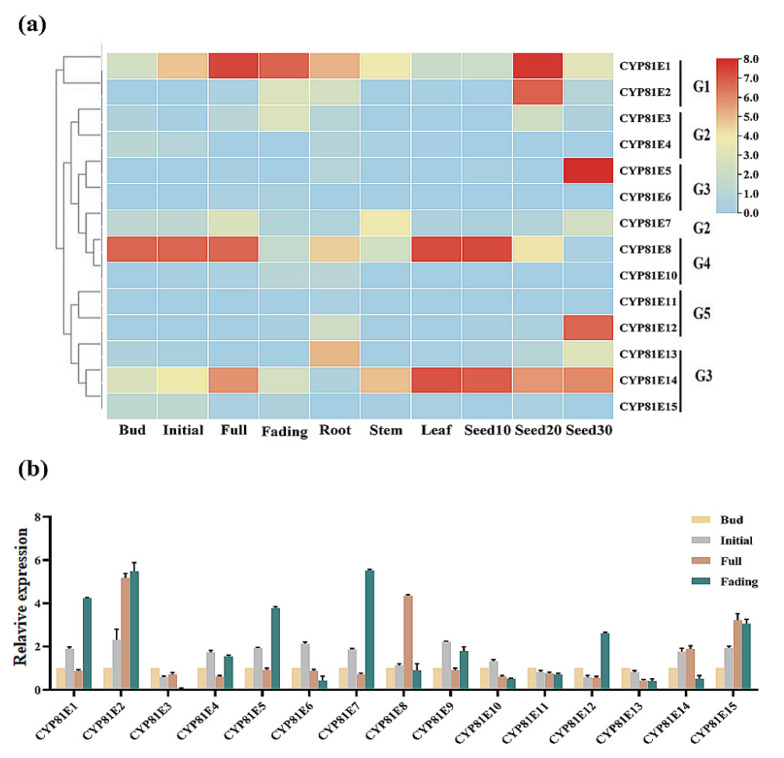
Expression analysis of CtCYP81E genes in safflower at different flowering stages. (**a**) the differential expression level of CtCYP81E genes using the FPKM values obtained from RNA-seq data. The expression data were illustrated as a heatmap in five different tissues. The extent of expression is shown by the color spectrum to the right, with red denoting strong expression and green denoting low expression. (**b**) The quantitative real-time expression variations of the CtCYP81E genes using qRT-PCR test in four flowering stages. The 18S rRNA gene was used to standardize the relative fold expression level, and the data were presented as means ± SE (*n* = 3).

**Figure 5 molecules-27-08001-f005:**
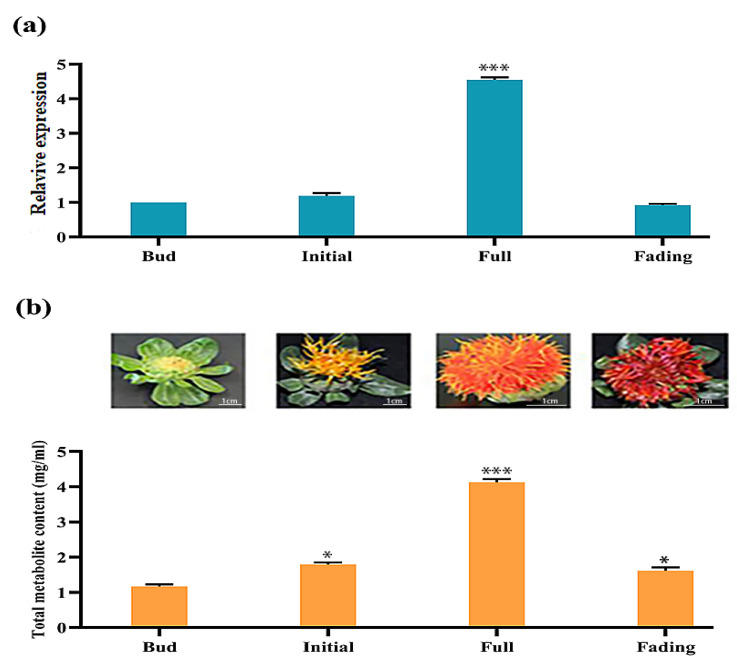
Expression The reciprocal relation of CtCYP81E8 expression and accumulation of total metabolites in four different flowering stages of safflower (**a**) the quantitative expression analysis of CtCYP81E8 using qRT-PCR assay (**b**) four different flowering stages of safflower and accumulation of hydroxysafflor yellow pigment. Data were presented as means ± SE (*n* = 3), and the asterisks * denotes *p* < 0.05 and *** denotes *p* < 0.001.

**Figure 6 molecules-27-08001-f006:**
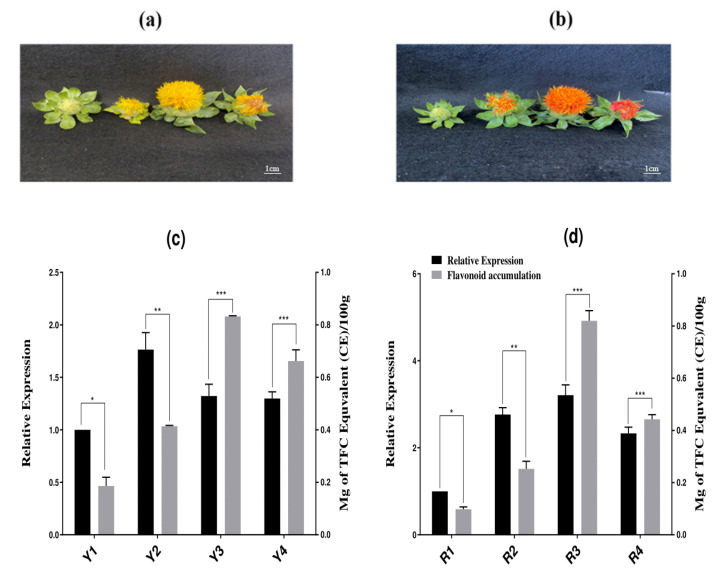
The correlation analysis of CtCYP81E8 expression and flavonoid accumulation in yellow and red-type safflower varieties. Phenotype of (**a**) yellow-type safflower and (**b**) red-type safflower (**c**) qRT-PCR expression analysis of CtCYP81E8 and total flavonoid accumulation in four different flowering stages of yellow type safflower (**d**) qRT-PCR expression analysis of CtCYP81E8 and total flavonoid accumulation in four different flowering stages of red-type safflower. Data were presented as means ± SE (*n* = 3), and the asterisks * denotes *p* < 0.05, ** denotes *p* < 0.01, and *** denotes *p* < 0.001. Alpha-numeric codes denote; Y: yellow type safflower and R: red-type safflower. Y1/R1: bud flowering, Y2/R2: initial flowering, Y3/R3: full flowering, Y4/R4: Fade flowering.

**Figure 7 molecules-27-08001-f007:**
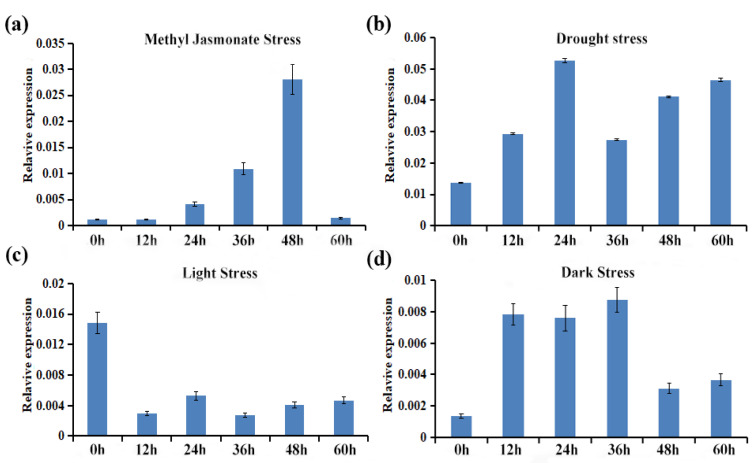
The effects of methyl jasmonate, drought, and light and dark stress on safflower CtCYP81E8 gene expression. Six separate time periods (0, 12, 24, 36, 48, and 60 h) of relative fold expression data were displayed on graphs. Three biological replicates were analyzed to determine the variation in expression level. Data were presented as means ± SE (*n* = 3).

**Figure 8 molecules-27-08001-f008:**
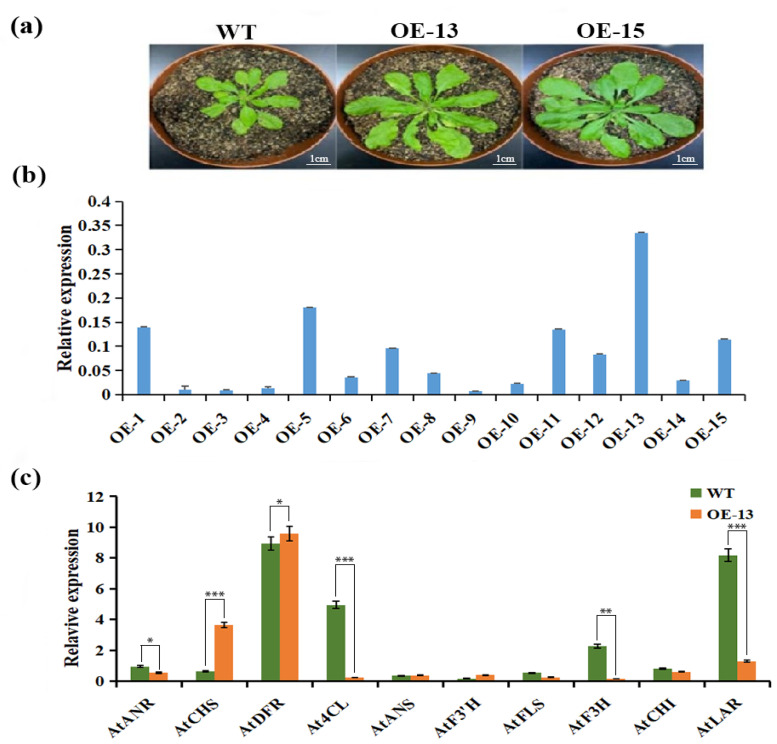
The relative expression analysis of CtCYP81E8 and core structural genes of flavonoid biosynthetic pathway in transgenic *Arabidopsis*. (**a**) The Phenotype of wild-type, OE-13, and OE-15 transgenic lines. (**b**) The expression level of CtCYP81E8 in different transgenic lines (OE1-15). (**c**) The expression level of key structural genes involved in the downstream regulatory pathway of flavonoid biosynthesis. Gene encoding these enzymes includes: Anthocyanidin reductase (ANR), CHS (Chalcone synthase), DFR (Dihydroflavonol 4-reductase), ANS (Anthocyanidin synthase), F3′H (flavonoid 3′-hydroxylase), FLS (Flavonol synthase), F3H (Flavonoid 3-hydroxylase), CHI (Chalcone isomerase) LAR, (Leucoanthocyanidin reductase). The relative fold expression level was normalized according to the 18S rRNA gene. Data were presented as means ± SE (*n* = 3), and the asterisks * denotes *p* < 0.05, ** denotes *p* < 0.01, and *** denotes *p* < 0.001.

**Figure 9 molecules-27-08001-f009:**
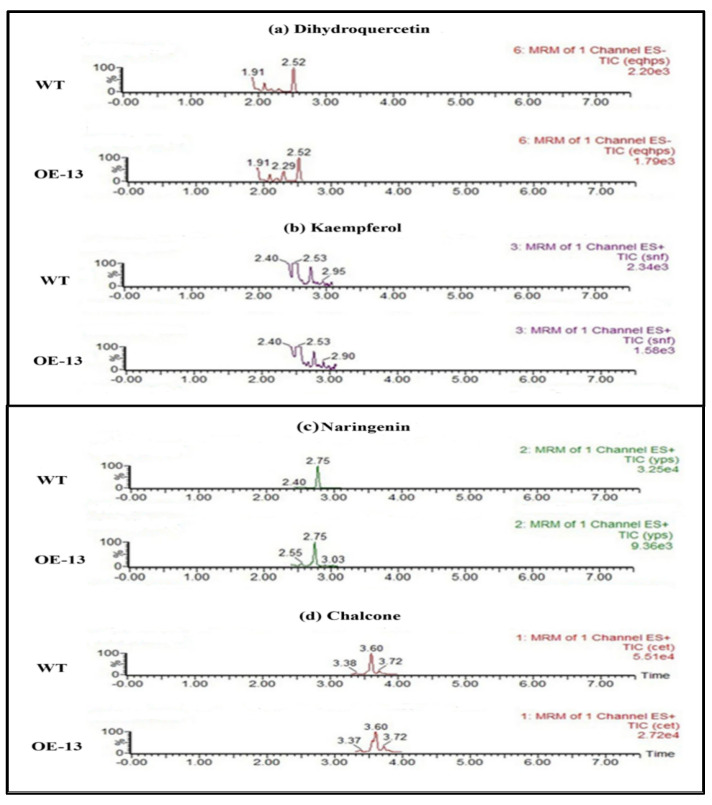
HPLC-MS/MS profiling of OE-13 transgenic line and WT *Arabidopsis*. The quantification of metabolite in transgenic *Arabidopsis* and WT plants. The numbers in abscissa represent: (**a**) 6: eqaps denotes dihydroquercetin; (**b**) 3: snf denotes kaempferol; (**c**) 2: yps denotes naringenin; and (**d**) 1: cet denotes chalcone.

**Table 1 molecules-27-08001-t001:** The physicochemical properties of safflower CYP81E subfamily proteins.

Gene Name	Gene ID	Protein Length	PI	MW	Subcellular Localization	IS Index	*Arabidopsis* Homology	GRAVY
CYP81E1	CCG011365.1	516	8.38	58.2	Plasma membrane	49.02	AT4G37370.1	−0.111
CYP81E2	CCG011366.1	501	8.96	57.29	Plasma membrane	46.06	AT4G37370.1	−0.125
CYP81E3	CCG011367.1	501	8.73	57.29	Plasma membrane	43.78	AT4G37370.1	−0.162
CYP81E4	CCG011368.1	339	5.61	38.28	Plasma membrane	40.66	AT1G66540.1	−0.332
CYP81E5	CCG007304.1	506	6.78	57.57	Plasma membrane	53.61	AT4G37370.1	−0.173
CYP81E6	CCG007305.1	506	6.78	57.57	Plasma membrane	53.61	AT4G37370.1	−0.173
CYP81E7	CCG007306.1	440	6.45	50	Plasma membrane	54.2	AT4G37370.1	−0.255
CYP81E8	CCG007309.1	337	6.49	38.48	Plasma membrane	44.14	AT1G66540.1	−0.32
CYP81E9	CCG007984.1	115	4.74	12.97	Plasma membrane	70.48	AT4G37370.1	−0.64
CYP81E10	CCG008862.1	246	9.35	27.85	Plasma membrane	47.65	AT4G37320.1	−0.147
CYP81E11	CCG008863.1	177	6.83	19.94	Endoplasmic reticulum	47.57	AT1G66540.1	−0.194
CYP81E12	CCG013393.1	435	8.49	49.49	Plasma membrane	44.98	AT4G37360.1	−0.143
CYP81E13	CCG013395.1	437	8.84	49.74	Plasma membrane	42.55	AT4G37340.1	−0.127
CYP81E14	CCG021962.1	516	7.68	59	Plasma membrane	45.29	AT4G37340.1	−0.156
CYP81E15	CCG004067.1	434	8.41	50.12	Plasma membrane	46.92	AT4G37370.1	−0.286

## Data Availability

Not Applicable.

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
