# Peer review of "Molecular Characterization of an Isoflavone 2′-Hydroxylase Gene Revealed Positive Insights into Flavonoid Accumulation and Abiotic Stress Tolerance in Safflower"

_molecules, 2022, doi:10.3390/molecules27228001_

Round 1

Reviewer 1 Report

Dear Authors

The research work entitles on “Identification and overexpression of safflower isoflavone 2’-hydroxylase regulates flavonoid biosynthesis and abiotic stress tolerance in transgenic Arabidopsis” was done and reported by Jianyu Liu et al.

The research work is so good and well explained. It was focused at molecular level and clearly discussed about the results. Over all the writing part is well.

But few of the figure quality is not accepted. Provide clear figures.

Thank you

Author Response

The research work entitles on “Identification and overexpression of safflower isoflavone 2’-hydroxylase regulates flavonoid biosynthesis and abiotic stress tolerance in transgenic Arabidopsis” was done and reported by Jianyu Liu et al.

The research work is so good and well explained. It was focused at molecular level and clearly discussed about the results. Over all the writing part is well.

But few of the figure quality is not accepted. Provide clear figures.

Thank you

Response: We are very grateful to the Reviewer for his encouragement and appreciation of our manuscript. We agree with the reviewer’s point of view about the quality of some figures. Following your comment, we have improvised the low-quality figures within the revised manuscript. The quality of the figures was thoroughly improved by maximizing the pixels of each figure. Please refer to the updated figures in the revised manuscript.

Reviewer 2 Report

In this study, authors report the structural and functional diversity, conserved topology and diverse expression pattern of CtCYP81E encoding genes from safflower. In addition, the function of a putative isoflavone 2’-hydroxylase in transgenic Arabidopsis was investigated. The spatiotemporal expression of CtCYP81E8 under methyl jasmonate, polyethylene glycol, light, and dark conditions further highlighted its likely significance in abiotic stress adaption.  

The following questions need to be answered before the article can be published. 

1、   The title needs to be changed.

2、   Table1 Line 111: The format of the table needs to be corrected. Three lines table format is necessary.

3、   Figure 1: The gene information in the phylogenetic tree construction should be provided as a supplementary table.

4、   Figure 4-7 In the Figures showing the gene expression quantity, the ordinate of the chart was not consistent. In the MS, several descriptions occurred, such as “the relative expression level” , “relative mRNA expression”,  and “relative expression” . In addition, statistical data should be given. It is better to give a significant difference in the graph.

5、   Figure 5b, Figure 6a, 6b and Figure 8a: the bar representing the size should be provided.

6、   For identifying the function of CYP81E, the gene editing for gene function loss is necessary.

7、   Figure9: The flavonoid content was determined in OE-13 using HPLC-MS/MS. The results from at least three lines should be provided.

8、   Please check that all supplementary materials, including the repeatability of raw data, missing headers and annotations for each table. Please note the font and font size is not consistent in the manuscript.

9、   There are some grammar mistakes throughout the manuscript, please check it.

10、The data is not enough to support that CtCYP81E is related to abiotic stress tolerance. Abiotic stress tolerance should be tested using transgenic Arabidopsis.

Author Response

In this study, authors report the structural and functional diversity, conserved topology and diverse expression pattern of CtCYP81E encoding genes from safflower. In addition, the function of a putative isoflavone 2’-hydroxylase in transgenic Arabidopsis was investigated. The spatiotemporal expression of CtCYP81E8 under methyl jasmonate, polyethylene glycol, light, and dark conditions further highlighted its likely significance in abiotic stress adaption.  

 The following questions need to be answered before the article can be published. 

 The title needs to be changed.

Response 1: Thank you for recommending amendments in the title. We have changed the title to a more compelling one during the revisions.

  • Table1 Line 111: The format of the table needs to be corrected. Three lines table format is necessary.

Response 2: The format of Table 1 has been corrected according to the journal’s standard format.

  • Figure 1: The gene information in the phylogenetic tree construction should be provided as a supplementary table.

Response 3: Thank you for this comment. We have supplied the gene and protein sequence information of all CtCYP81E members used in the phylogenetic tree in the supplementary file as Table S2 and S3.

  • Figure 4-7 In the Figures showing the gene expression quantity, the ordinate of the chart was not consistent. In the MS, several descriptions occurred, such as “the relative expression level” , “relative mRNA expression”,  and “relative expression” . In addition, statistical data should be given. It is better to give a significant difference in the graph.

Response 4: Thank you for pointing out this lapse. Following your comment, we have corrected the description of the ordinate used in most of the figures where applicable to a consistent one ‘‘Relative expression’’ in the revised manuscript. The statistical significance has been added to each bar in these figures. Please refer to the revised Figures ordinate (Fig4-8) in the revised manuscript.

  • Figure 5b, Figure 6a, 6b and Figure 8a: the bar representing the size should be provided.

Response 5: We are very thankful to the reviewer for his constructive comments and suggestions. According to your suggestion, we have added the scale bar on Figure 5b, Figure 6a, 6b and Figure 8a during the revisions. Please refer to the revised Figure 5b, Figure 6a, 6b and Figure 8a in the revised manuscript.

  • For identifying the function of CYP81E, the gene editing for gene function loss is necessary.

Response 6: We agree with the reviewer’s point of view about the CYP81E loss of function study via gene editing. However, due to the lack of enough funding we could not manage to investigate this part. At this point, we rely on the functional characterization of CYP81E using a genome-wide strategy followed by expression analysis in safflower and transgenic plants and then compared its potential role in flavonoid accumulation. We aimed to provide a valuable reference resource and narrow down the genetic toolbox for future studies related flavonoid biosynthesis and abiotic stress resistance strategies by utilizing gene editing technologies in the future.

  • Figure9: The flavonoid content was determined in OE-13 using HPLC-MS/MS. The results from at least three lines should be provided.

Response 7: Thank you for your suggestion. We agree about your concern about the result of HPLC-MS/MS in OE-13 transgenic line. The reason why we choose only OE-13 line for the determination of flavonoid content via HPLC-MS/MS is because of its high expression out of all transgenic lines described in Fig.8b. Similarly, upon confirmation of induced expression level of other core flavonoid pathway genes, we assumed that the increased relative fold expression of CtCYP81E8 might have positive correlation with the accumulation of flavonoid content in comparison to other transgenic lines.

  • Please check that all supplementary materials, including the repeatability of raw data, missing headers and annotations for each table. Please note the font and font size is not consistent in the manuscript.

Response 8: We have corrected these issues in the supplementary files during the revision. Please refer to the revised supplementary files with the revised manuscript.

  • There are some grammar mistakes throughout the manuscript, please check it.

Response 9: Thank you for highlighting this lapse. We have corrected all grammatical issues within the revised manuscript.

10、The data is not enough to support that CtCYP81E is related to abiotic stress tolerance. Abiotic stress tolerance should be tested using transgenic Arabidopsis.

Response 10: Thank you for highlighting this point. We are sorry for the previous misleading title as rightly pointed out by another reviewer as well. The abiotic stress tolerance assay was only investigated in safflower; therefore, we have corrected the main title of the manuscript on the basis of the results presented, thereafter. Please refer to the revised title for the better implication of the results and data presented in this manuscript.

Reviewer 3 Report

Line 392-415, the authors tried to picture CtCYP81E8 gene expression with accumulation of flavonoid in tissue at different stage. The relative gene expression in all stages completely is less than 0.1, which means downregulated, question is  which stage was used for comparing ? as said in M&M, 2-∆∆Ct method was used an qPCR data analysis, if so, relative gene expression in one of four stages in data analysis in  qPCR should be 1.

Line 436, Figure 7, not sur why authors used different way presenting the expression of gene, instead of  2-∆∆Ct method.

line 729-785, these three section could be combined, some of contents repeatedly presented in three sections.

line 757 ‘2Ct approach’, what is this ? you mean 2- ∆∆Ct ?

line 760, please add the citation of method used for metabolite analysis.

Line 790-791  ‘300 mmol/L’ is concentration, how much of this solution was applied to each plant each time ?

Line 802-810 please clarify if the sequence of CtCYP81 used here is cDNA, not sure ‘the genomic sequence of CtCYP81E8’ mentioned here is genomic sequence or mRNA/cDNA.

In Figure 9. You have a-d panels, please make the figure legend consistent with figure, pay attention to letter used in figure, uppercase or lowercase, need to be consistent.

In overall data analysis, please give p value and what is statistical method used in data analysis in M&M,  

Author Response

Line 392-415, the authors tried to picture CtCYP81E8 gene expression with accumulation of flavonoid in tissue at different stage. The relative gene expression in all stages completely is less than 0.1, which means downregulated, question is  which stage was used for comparing ? as said in M&M, 2-∆∆Ct method was used an qPCR data analysis, if so, relative gene expression in one of four stages in data analysis in  qPCR should be 1.

Response 1: we are very thankful to the reviewer for pointing out this lapse. Following your comment, we have re-analyzed the expression level of  CtCYP81E8 gene in different stages of red and yellow flowering by normalizing the expression level using the 2-∆∆Ct method. The Bud flowering stage was used for comparison. The figure 6 has been corrected in the revised manuscript. Please refer to the revised figure 6.

Line 436, Figure 7, not sur why authors used different way presenting the expression of gene, instead of  2-∆∆Ct method.

Response 2: The expression level of CtCYP81E8 was investigated under different stress conditions at different timepoints. Here, we specifically compared the actual fold change level with 0 timepoint different stress conditions were used thus we only compare actual expression level of 0h with other timepoints.

line 729-785, these three section could be combined, some of contents repeatedly presented in three sections.

Response 3: Thank you for your good suggestions. As section 4.4 described the expression level of 15 CtCYP81E encoding genes in different flowering stages altogether. We believe that this section could not combined with the following sections. However, the other two sections were combined according to your suggestion within the revised manuscript. Changes to the manuscript were highlighted with yellow color font.

line 757 ‘2Ct approach’, what is this ? you mean 2- ∆∆Ct ?

Response 4: We are sorry for this typo error. Here it means 2- ∆∆Ct  which has been corrected in the revised manuscript.

line 760, please add the citation of method used for metabolite analysis.

Response 5: Thank you reminding this. We have added a citation of the methods used for the metabolite analysis in the revised manuscript. Please refer to the newly added reference in the revised manuscript.

Line 790-791  ‘300 mmol/L’ is concentration, how much of this solution was applied to each plant each time ?

Response 6: The exact volume of this MeJA concentration which was sprayed was 200 mL.

Line 802-810 please clarify if the sequence of CtCYP81 used here is cDNA, not sure ‘the genomic sequence of CtCYP81E8’ mentioned here is genomic sequence or mRNA/cDNA.

Response 7: The sequence of CtCYP81 used here is cDNA . We have corrected this point in the revised manuscript.

In Figure 9. You have a-d panels, please make the figure legend consistent with figure, pay attention to letter used in figure, uppercase or lowercase, need to be consistent.

Response 8: Thank you for highlighting this lapse. We have corrected the figure legend and letter consistency in Fig.9 within the revised manuscript. Please see the modified legend and figure 9 in the revised manuscript.

In overall data analysis, please give p value and what is statistical method used in data analysis in M&M,  

Response 9: Thank you for your good suggestion. Following your comment, we have added a separate section of 2.10: Statistical Analysis in M&M section in the revised manuscript. In addition, the p-value was added to each data analysis used in this manuscript. Changes to the modified manuscript were highlighted with yellow text. 

Round 2

Reviewer 2 Report

The MS was significantly improved. However, the following questions should be concerned before it can be published.

1 TitleIt would be better to change the title to “Molecular characterization of an isoflavone 2’-hydroxylase gene revealed positive insights into flavonoid accumulation and abiotic stress tolerance in safflower”

2 Figure 4: Figure legend: the data were presented as means ± SE (n=3) with a p-value (p ≤ 0.05). P- value is used for the significant difference comparison between/among samples. In Figure 4, I did not see any comparison. It can be changed to “the data was presented as means ± SE (n=3)”.

3 Figure 5: Data were presented as means ± SE (n=3) with a p-value (p ≤ 0.05). Usually different P- values were used for different significant difference description. For example, * P≤ 0.05, ** P≤ 0.01 etc. Please revised your statistical description.

4 Figure 6:The data were presented as means ± SE (n=3) with a p-value (p ≤ 0.05). Please revised your statistical description.

5 Figure 7: the data were presented as means ± SE (n=3) with a p-value (p ≤ 0.05).

Please revised your statistical description.

6 Figure 8: Please revised your statistical description.

7 line 671: Solanum Lycopersicum, should be Solanum lycopersicum

8 line 835: EcoRI, R1 should be not italic.

9 line 836: BamHI, H1 should be not italic.

Author Response

We would like to thank the reviewer for suggesting minor revisions on our manuscript. Following these suggestions, we have made minor changes to the manuscript. The reviewer comments are reproduced here with our point-by-point responses in red color text. Changes to the manuscript were highlighted with yellow color font. 

Comments and Suggestions for Authors

The MS was significantly improved. However, the following questions should be concerned before it can be published.

1 Title:It would be better to change the title to “Molecular characterization of an isoflavone 2’-hydroxylase gene revealed positive insights into flavonoid accumulation and abiotic stress tolerance in safflower”

Response 1:  The main title of the manuscript is change in accordance with your suggestions. 

2 Figure 4: Figure legend: the data were presented as means ± SE (n=3) with a p-value (p ≤ 0.05). P- value is used for the significant difference comparison between/among samples. In Figure 4, I did not see any comparison. It can be changed to “the data was presented as means ± SE (n=3)”.

Response 2:  We apologise for this lapse. The fig. 4 legend has been corrected in the revised manuscript. 

3 Figure 5: Data were presented as means ± SE (n=3) with a p-value (p ≤ 0.05). Usually different P- values were used for different significant difference description. For example, * P≤ 0.05, ** P≤ 0.01 etc. Please revised your statistical description.

Response 3:  Thank you for highlighting this point. We have correctred this issue by adding a statistcal description in Fig.5 legend (Data were presented as means ± SE (n=3), and the asterisks * denotes P <0.05, ** denotes P <0.01 and *** denotes P <0.001). Changes to the manuscript were highlighted in yellow color text in the revised manusript.  

4 Figure 6:The data were presented as means ± SE (n=3) with a p-value (p ≤ 0.05). Please revised your statistical description.

Response 4:  We have revised the statement of the statistcal description in Fig.6 legend (Data were presented as means ± SE (n=3), and the asterisks * denotes P <0.05, ** denotes P <0.01 and *** denotes P <0.001). Changes to the manuscript were highlighted in yellow color text in the revised manusript.  

5 Figure 7: the data were presented as means ± SE (n=3) with a p-value (p ≤ 0.05).

Please revised your statistical description.

Response 5:  We have revised the statement of the statistcal description in Fig.7 legend (Data were presented as means ± SE (n=3), and the asterisks * denotes P <0.05, ** denotes P <0.01 and *** denotes P <0.001). Changes to the manuscript were highlighted in yellow color text in the revised manusript. 

6 Figure 8: Please revised your statistical description.

Response 6:  We have revised the statement of the statistcal description in Fig.8 legend (Data were presented as means ± SE (n=3), and the asterisks * denotes P <0.05, ** denotes P <0.01 and *** denotes P <0.001). Changes to the manuscript were highlighted in yellow color text in the revised manusript. 

7 line 671: Solanum Lycopersicum, should be Solanum lycopersicum

Response 7:  Thank you for the correction. We have corrected this point in the revised version. 

8 line 835: EcoRI, R1 should be not italic.

Response 8: We have corrected this point in the revised version. 

9 line 836: BamHI, H1 should be not italic.

Response 9:  We have corrected this point in the revised version.